# An aperiodic chiral tiling by topological molecular self-assembly

Jan Voigt[1], Miloš Baljozović [1], Kévin Martin[2], Christian Wäckerlin[3,4], Narcis Avarvari [2] ✉ & Karl-Heinz Ernst [1,5,6] ✉

Studying the self-assembly of chiral molecules in two dimensions offers insights into the fundamentals of crystallization. Using scanning tunneling microscopy, we examine an uncommon aggregation of polyaromatic chiral molecules on a silver surface. Dense packing is achieved through a chiral triangular tiling of triads, with $N$ and $N \pm 1$ molecules at the edges. The triangles feature a random distribution of mirror-isomers, with a significant excess of one isomer. Chirality at the domain boundaries causes a lateral shift, producing three distinct topological defects where six triangles converge. These defects partially contribute to the formation of supramolecular spirals. The observation of different equal-density arrangements suggests that entropy maximization must play a crucial role. Despite the potential for regular patterns, all observed tiling is aperiodic. Differences from previously reported aperiodic molecular assemblies, such as Penrose tiling, are discussed. Our findings demonstrate that two-dimensional molecular self-assembly can be governed by topological constraints, leading to aperiodic tiling induced by intermolecular forces.

Dating back to ancient times, the art of tiling has been an integral part of human society, with well-known instances including the mosaics of Roman and Arabic cultures[1]. Over the past century, aperiodic sequences of their building blocks have been identified in nucleic acids, such as DNA, for example, and proteins. However, aperiodic two-dimensional tilings have primarily played a significant role in mathematics[2]. A pattern is labeled aperiodic when it lacks a description through unchanging translational vectors. Illustrative two-dimensional examples include quasicrystals and Penrose tiling[3]. The emergence of Wang-Tiles, driven by challenges in computer science, has proved influential[4], even finding application in the design of artificial DNA assemblies[5]. Geometers are in constant pursuit of innovative tessellation patterns, with recent advancements yielding an aperiodic tiling originating from a single polygon[6], which has to break mirror symmetry[7]. These arrangements can offer potential applications in

materials science, particularly in the development of materials with enhanced electronic, optical, and mechanical properties, due to the precise control over structural patterns without the constraints of periodicity. For example, a recent theoretical study predicted exceptional physical properties for such aperiodic chiral two-dimensional (2D) material[8]. In this study, we demonstrate that the two-dimensional molecular self-assembly of a chiral aromatic hydrocarbon on a silver surface can lead to a chiral aperiodic tiling. Each single molecule serves as a symmetric chiral monotile, allowing for an aperiodic tiling of the plane at the supramolecular level.

Chirality, a concept ubiquitous across natural sciences including chemistry, physics, biology, and physiology[9,10], holds immense significance. The crystallization of chiral molecules has played a pivotal role in comprehending molecular structures[11–13] and remains a cornerstone method for achieving enantiopure pharmaceuticals and

[1]Empa, Swiss Federal Laboratories for Materials Science and Technology, Dübendorf, Switzerland. [2]Univ Angers, CNRS, MOLTECH-Anjou, SFR MATRIX, F-49000 Angers, France. [3]Laboratory for X-ray Nanoscience and Technologies, Paul-Scherrer-Institut (PSI), CH-5232 Villigen PSI, Switzerland. [4]Institute of Physics, Swiss Federal Institute of Technology Lausanne (EPFL) Station 3, CH-1015 Lausanne, Switzerland. [5]Nanosurf Lab, Institute of Physics of the Czech Academy of Sciences, Prague, Czech Republic. [6]Department of Chemistry, University of Zürich, Zürich, Switzerland. ✉e-mail: narcis.avarvari@univ-angers.fr; kalle@fzu.cz

fragrances[14]. The aggregation of a balanced mixture of left- and right-handed molecules can result in enantiopure but opposite-handed crystal conglomerates or crystals in which both enantiomers populate the crystal unit cell (racemate crystal). In rare instances, both enantiomers are randomly distributed within the crystal lattice, forming a solid solution[15,16]. Despite being observed for more than 175 years[17], predicting the outcome of chiral crystallization remains elusive. To gain insight into complex material phenomena, model systems with high control and reduced complexity are invaluable. Consequently, the exploration of chiral molecule aggregation on surfaces using scanning tunneling microscopy (STM) has proven instrumental in enhancing our understanding of chiral molecular recognition[15].

## Results

### Topological self-assembly

An important class of chiral aromatic chemical compounds are helicenes[18]. The helical arrangement of electronic states resulting from their *ortho*-annelated aromatic rings renders them highly promising for applications like chiral photonics and chirality-based spintronics[19–23]. In contrast to the rigid helicenes employed in previous 2D aggregation studies[24], tris(tetrahelicenebenzene) (t[4]HB; 1,3,5-tris(benzo[c]phenanthren-2-yl)benzene, Fig. 1a) boasts greater flexibility. Its structure comprises three tetrahelicene arms connected via single bonds to a central benzene unit (see Methods for the preparation and characterization). The conformers with the lowest energy all have three arms with identical handedness. The helicity can be either left-handed, denoted as *M* (minus) with the arm spiraling counterclockwise from tip to center away from the observer, or right-handed, denoted as *P* (plus) with a clockwise spiral. The molecule's flexibility stems from the low barriers of helix inversion in the helical arm units (3.5–4.4 kcal mol$^{-1}$)[25,26], and the potential rotation of arms around the three carbon–carbon single bonds.

Figure 1b shows an STM image recorded from a densely packed monolayer of t[4]HB on Ag(111) at 120 K subsequent to deposition under ultra-high vacuum conditions at room temperature (see "Methods"). The area shown in Fig. 1b contains predominantly *(M)*-t[4]HB. The distribution of the minority *(P)*-enantiomer in Fig. 1b is shown in Supplementary Fig. 1. A corresponding STM image of a mirror domain with predominantly *(P)*-t[4]HB molecules is shown in Supplementary Fig. 2.

Notably, the 2D self-assembly here shows different topological defects (termed 'nodes' here). Supramolecular spirals with two and three arms originate from two different defects (2- and 3-nodes, respectively). There exist zero-nodes that do not serve as spiral origins but have a void at their center. The 3-nodes also show a void while the 2-nodes, being overcrowded, cause the arms of connecting molecules to bend partly away from the surface. The topology of different nodes can be indexed by a procedure determining the displacement in molecular unit vectors of the supramolecular spirals (Fig. 1c). For the rules of the procedure encircling the structure along the white line see Supplementary Fig. 3. Specifically, 2-node and 3-node structures prompt displacements of two and three molecular unit vectors, respectively (Fig. 1c, indicated by yellow arrows). Given that the monolayer also breaks mirror symmetry, these 0-, 2- and 3-nodes exist globally in two enantiomorphous forms (Fig. 1c). Detailed structural models of the molecular arrangement around nodes are illustrated in Fig. 1d.

STM images single molecules as triskels (inset of Fig. 1e) where the tips of the three arms appear brighter. As a result, the helical arm tips are positioned slightly higher above the surface, while the remaining parts of the molecule maintain a parallel orientation to the surface (inset Fig. 1f). This conformation ensures optimal van der Waals contact with the surface, as only the three terminal benzo groups at the tips are elevated. Furthermore, all three arms possess the same helical sense, simplifying the determination of the molecule's handedness,

even without high-resolution STM height data. Contrastingly, deposition onto the surface at low temperatures shows disordered layers and many molecules in meta-stable Y-configurations (Supplementary Fig. 4). Upon thermal activation, however, all molecules transition into a state of defined homochirality, highlighting the relatively high flexibility due to the low energy barrier required for switching handedness and rotating the arms around the single C–C bond.

In addition to nodes, the molecular monolayer exhibits triangles characterized by hexagonally arranged molecules, as depicted in Fig. 1e, f. Within a triangle, the molecules undergo a 60° rotation compared to molecules in adjacent triangles. At the boundary between triangles, molecules are packed slightly denser due to the relative 60° rotation, which permits a closer interdigitation of the helical arms. Such packing strategy, in which a newly arriving molecule either maintains the hexagonal growth or starts a new hexagon by creating a boundary, has been previously reported for achiral species on surfaces[27,28]. A detailed evaluation of the energetic conditions at such boundary has been performed for a triangular-shaped species but completely ignored entropic aspects[29]. The structural models of the boundaries between triangles primarily containing either *(M)*- or *(P)*-enantiomers are presented in Fig. 1g. Molecules in neighboring triangles are also situated above distinct adsorption sites. For instance, if molecules in one triangle are positioned with their central benzene ring over a hexagonally closed-packed (hcp) site on the Ag(111) surface, in adjacent triangles, they will be situated above face-centered cubic (fcc) threefold hollow sites. Notably, only the relative arrangement and not the exact adsorbate site is ascertainable here. However, due to the adsorption site displacement between triangles, the unit cell of the boundary (and thus the entire molecular lattice) becomes incommensurate with the underlying Ag(111) lattice.

The self-assembly process of t[4]HB on Ag(111) showcases an intriguing feature: the triangles can manifest in various sizes within a single domain but encompassing only triangles of $N \pm 1$. Here, $N$ represents the number of molecular unit vectors forming the triangular boundary. In the structure depicted in Fig. 1e, $N = 5$ predominates, while $N = 4$ and $N = 6$ are less common. Similarly, the example from the 'mirror world' in Fig. 1e exhibits a predominant $N = 11$, along with minor occurrences of $N = 10$ and $N = 12$. Overall, domains showcasing $N$ values ranging from 2 to 15 are also observed (Supplementary Fig. 5), either after fresh sample preparation or when the STM tip is repositioned across the sample at a considerable distance of a few millimeters. It is important to emphasize that these different results have been obtained under identical preparation conditions.

### Chirality transfer

For the ensuing discussion on tiling and the connection between supramolecular nodes and triangular subdomains, it proves beneficial to map colored triangles onto the molecular assembly (Fig. 2). This abstraction effectively highlights the correlation between molecular handedness and node handedness, as well as the association between the predominant triangle size $N$ and the adjacent $N-1$ and $N+1$ size triangles. Initially, a color code designates the enantiomer and its azimuthal orientation (Fig. 2a), which is then applied to the triangular subdomain as colored triangles, each with the corresponding $N$ value (Fig. 2b). Given the inherent 60° rotation relationship between adjacent triangles and the separation of mirror domains, the color code can be omitted without sacrificing generality. In addition, a simplified molecular stick model accentuates the interlocking helicene arms at the boundary (Fig. 2c, top). As the molecular handedness dictates the direction of displacement, triangles can only shift either to the left or the right. Consequently, the transmission of chirality from the individual molecule to the mesoscopic scale is facilitated by the prescribed displacement at triangle boundaries.

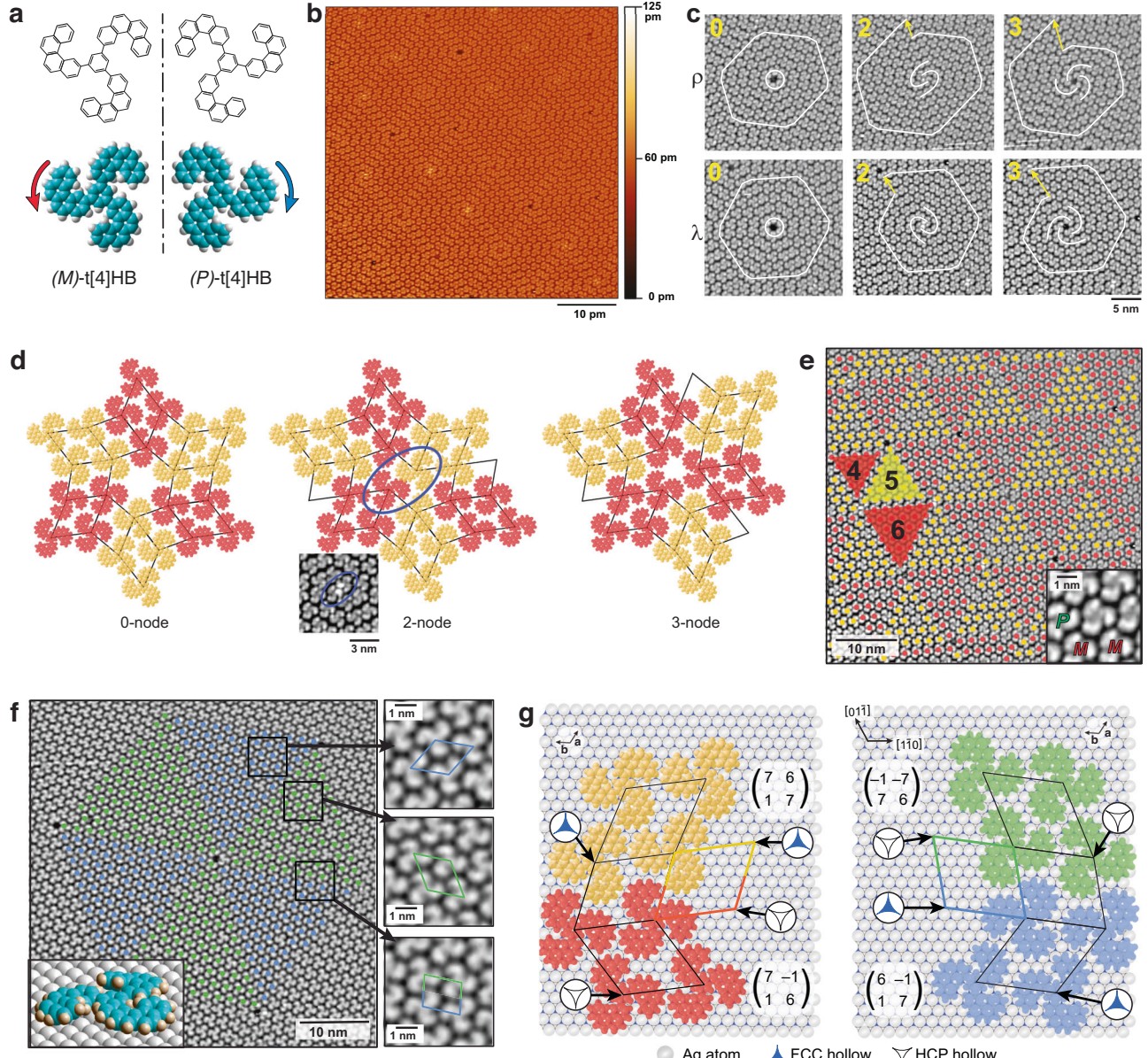

**Fig. 1 | Self-assembly in two-dimensions containing topological defects.**
**a** Structure models of both enantiomers of t[4]HB. In (M)-t[4]HB the three arms spiral away from the observer counterclockwise, in (P)-t[4]HB clockwise. **b** STM image of monolayer of t[4]HB on Ag(111) showing an arrangement of supramolecular triangles and spiral nodes. The range of height probed is represented by the color scale bar. **c** Classification of observed nodes in both mirror worlds. Besides a zero-node with a void in the center, 2-armed and 3-armed spiral arrangements exist, which show after one turn an offset of 2 or 3 molecular lattice vector units (yellow arrows), respectively. ρ-nodes occur in mirror domains with (P)-t[4]HB as majority, mirror domains with λ nodes have (M)-t[4]HB as majority. Scale bar applies to all six STM images. **d** The molecular arrangements of (M)-enantiomers around the nodes show either voids (0- and 3-node) or overcrowding (2-node). Yellow and red colored molecules differ by 60° rotation. Unit cells of the hexagonal arrangement in the triangles and the boundaries are indicated as parallelograms. The STM image for the 2-node suggests that two tetrahelicene arms of two molecules are bent upward. **e** STM image of a mirror domain containing predominantly (M)-t[4]HB.

Molecules marked with yellow dots are rotated clockwise by 60° with respect to molecules marked with red dots. Molecules without color-mark are (P)-t[4]HB enantiomers. Different triangle sizes are indicated with numbers for three examples. The inset shows both enantiomers in a single domain with submolecular resolution. The tips of helical arms are further above the surface. **f** STM image of a mirror domain containing predominantly (P)-t[4]HB. Molecules marked with green dots are rotated clockwise by 60° with respect to molecules marked with blue dots. The inset shows a model of a single (P)-adsorbate. The arrangements of the molecules in the different supramolecular triangles and at their boundary between both triangles are shown on the right. **g** The structure models of the boundaries between hexagonally ordered triangles in both mirror domains show interdigitation. Within the triangles, the molecules are arranged hexagonally. The periodicity of the molecular lattice with respect to the underlying Ag(111) lattice is given in matrix notation. The molecules of adjacent triangles are placed on different adsorption sites, either on top of hcp or fcc threefold hollow sites.

Three distinct categories of triangle interdigitation emerge. Equally sized triangles necessitate an inevitable offset contingent on the enantiomer's handedness (Fig. 2c, top). Furthermore, equally sized triangles may exhibit an additional offset of one molecular unit vector (Fig. 2c, middle), leading to a total offset of 2-unit vectors at that boundary. Lastly, combining triangles of different sizes, $N$ and $N \pm 1$, consistently results in an offset of one molecular unit vector (Fig. 2c, bottom). This framework enables a total of 10 different triangular pairing configurations (see Supplementary Fig. 6). Nodes with a topological index of 0, 2, and 3 will, respectively, feature total offsets

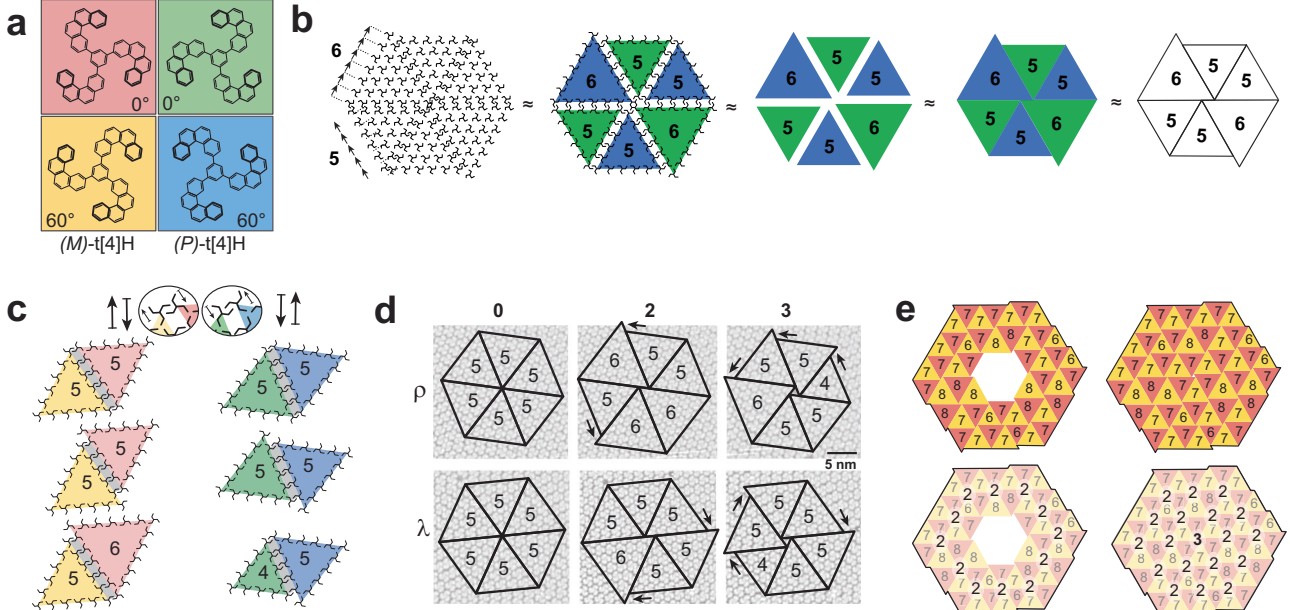

**Fig. 2 | Abstraction of molecular self-assembly into triangular offset-tiling.**
**a** Color code for both enantiomers of t[4]HB and their relative azimuthal orientation. **b** Presentation of a self-assembled pattern of t[4]HB *(P)*-enantiomers as triangles. The color stands for the azimuthal orientation as defined in (**a**) while the number stands for the relative size *N* of the triangle. **c** Modes of interdigitation between triangles at their common boundary. The handedness of the majority enantiomer defines the direction of the interdigitation offset of triangles (see ellipses). An additional offset by one molecular unit vector is also observed for equally sized triangles (middle) and for *N*±1 pairs (bottom). **d** Superposition of STM images of the different node types in both mirror domains with the abstraction of triangles. The node type is defined by the number of offsets as indicated by arrows. **e** Example of topological frustration. An area initially built by 2-nodes (*N* = 7±1) can only become completely filled if a 3-node is considered.

of 0, 2, and 3 (Fig. 2d). The topology established by these offsets translates into the spiral pattern when constructing nodes using *N*, *N* ± 1 triangles. Consequently, the handedness of supramolecular nodes emerges from a chiral displacement within the triangular context. It is important to note that deviations from these displacement rules have not been observed. Only 15 of many possible combinations of triangles within the *N*, *N* ± 1 scheme have been verified through STM observations (Supplementary Fig. 7). However, we could not identify any tiling rules that would forbid involvement of unobserved combinations.

To maintain 2D close packing here, topological strategies are at work. That is, the system adapts by either varying the triangle size by one molecular vector unit (*N* ± 1) or the node type. An illustrative example is presented in Fig. 2e. A 7 ± 1 triangle arrangement containing exclusively 2-nodes does not lead to close-packing. Subsequently, the central area mandates the inclusion of a 3-node structure to effectively cover the entire region.

### Entropy maximization

As no polar groups are present in the molecules, the lateral intermolecular interaction is dominated by van der Waals forces. Hence, the largest energy term involved in this self-assembly is the adsorption energy, i.e., the interaction energy between a single molecule and the silver surface. Using previous density functional theory calculations for polyaromatic hydrocarbons on gold as estimate, the binding might be as strong as 3.5 eV[30]. Consequently, the molecular layer strives for maximum density, setting this study apart from observations typically made in porous structures with directed intermolecular forces[31–34]. Compared to other metal surfaces, Ag(111) offers relatively high mobility, meaning that both lateral intermolecular forces and the shape of the molecules significantly influence the 2D self-assembly.

A good presentation of the relative energy involved is the lateral density (or local coverage). That is, the number of silver atoms located below a molecule should become as small as possible. Nodes with

voids in the center, such as 0- and 3-nodes are therefore energetically less favorable. On the other hand, a large number of nodes also increases the relative contribution of slightly denser boundaries between triangular domains. Therefore, from a certain triangle size on the disadvantage of uncovered node centers becomes compensated. Basically, at *N* ≥ 4, every molecule covers about 43 silver atoms, which is exactly the value for the pure hexagonal arrangement within a triangle. Consequently, for *N* ≥ 4, the energy landscape is flat across different triangular assemblies, suggesting that the 2D self-assembly is also influenced by entropy[31,35,36]. This entropic contribution explains the observation of a wide variety of structures with varying triangle sizes, even when the self-assembly is repeated under identical experimental conditions.

The chiral self-assembly of t[4]HB exhibits another distinctive character that increases entropy: the enantiomeric composition within single mirror domains is neither racemic nor homochiral. Instead, a solid solution predominantly composed of either the *(M)*- or the *(P)*-enantiomer (Fig. 1d and Fig. 1e, respectively) is observed. The enantiomeric excess (ee), calculated using the formula: ee = ([*M*] − [*P*])/([*M*] + [*P*]) for the area displayed in Fig. 1b and Supplementary Fig. 1, is determined to be 80%. Molecular mechanics calculations reveal that the incorporation of the 'incorrect' enantiomer is only disfavored by a mere 2 kcal mol⁻¹ (Supplementary Fig. 8). Hence, random incorporation of a minority enantiomer does not impose a notable energy penalty. However, it does contribute to the entropy of the aggregation process by $S = RT \ln 2$ for a single molecule, while for an ensemble of molecules the entire number of possible arrangements has to be taken into account (Suppl. Eq. (1)).

In some cases, the triangle size *N* exceeds the scan range of our STM and only a single domain of hexagonally ordered molecules without any nodes and triangle boundaries is observed (Fig. 3a). Interestingly, such purely hexagonally ordered domain contains typically more minority enantiomers (Fig. 3a, ee = 64%) than structures with nodes and boundaries. Such trend is confirmed by decreasing

## a

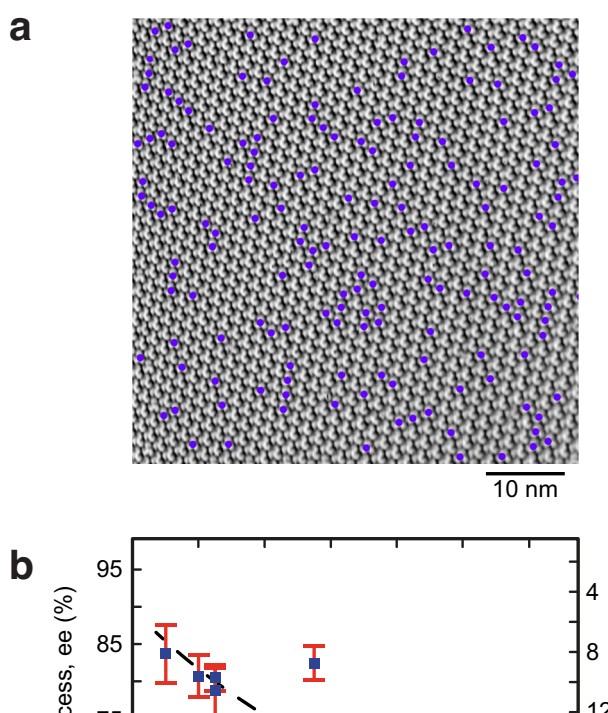

10 nm

## b

**Fig. 3 | Enantiomeric excess of a solid solution. a** STM image of a hexagonally ordered domain without nodes and triangle boundaries. Violet dots mark the minority *(M)*-enantiomers. **b** Dependence of enantiomeric excess with triangle size, as determined from STM images shown in Supplementary Fig. 5. The data represented in Fig. 3b originates from six independent measurements. In the statistical analysis, the chiral minority-majority occurrence was treated as a binary state problem with two outcomes. Hence, the error bars represent Standard Errors of the Mean from the binomial distribution. All presented data points demonstrate 99% significance as tested towards the hypothesis of pure racemic arrangement with 50:50 majority/minority content.

enantiomeric excess with increasing triangle size (i.e., less numbers of nodes per unit area, see Fig. 3b). This means that a lower entropy due to lower numbers of nodes becomes compensated by larger content of minority enantiomers (i.e., lower enantiomeric excess). At initial deposition ee is 0 but becomes automatically adjusted to the triangle size during growth.

A frozen lattice inherently lacks the capacity to augment mixing or configurational entropy. Consequently, the establishment of molecular surface dynamics becomes imperative during its formation. Deposition at either low coverage or low temperature fails to yield ordered aggregates. Under low coverage conditions, molecules maintain a considerable distance from one another within a temperature range spanning from room temperature down to 10 K (Supplementary Fig. 9). This observation signifies the presence of minute attractive interactions, suggesting that 2D self-assembly is primarily governed by repulsion at monolayer saturation coverage. The maximization of entropy and topological selection mandates dynamic processes within the lattice during growth, such as enantiomer

conversion and lateral rearrangement at a close-packed molecular layer. STM images acquired at room temperature for t[4]HB on Ag(111) indeed exhibit both long-range order and significant dynamics (Supplementary Fig. 10a). Electron diffraction further confirms the presence of order even at temperatures up to 80 °C (Supplementary Fig. 10b, c). Under these conditions, one should anticipate dynamic processes, such as site exchange and lattice healing, potentially involving second-layer sites. Notably, for helicenes, existing reports have detailed chiral crystallization dynamics occurring around full monolayer coverages[37,38].

Topological defects can be also introduced by an additive that fits well into the void of a 0-node, which compensates the energy penalty of a void. Hexaphenylbenzene (HPB), for example, fits nicely into the 0-node-void, hence changing the energy landscape and thus breaking the entropy-driven self-assembly (Fig. 4a). Doping the t[4]HB monolayer with HPB leads to a regular lattice structure. Depending on the amount of HPB added, its molecules are surrounded by different numbers of rings of t[4]HB molecules (Fig. 4b). The molecules within the same ring are rotated by 60° relative to one another, resulting in the densest packing formation. This self-assembly is primarily influenced by the maximum gain in adsorption energy achieved through dense packing. Below a additive fraction of 3% HPB, the self-assembly process continues to be driven by entropy.

## Discussion

What makes topological self-assembly different from regular self-assembly? In topological self-assembly additional constraints are imposed onto the assembly process. For example, topological defects (e.g., points, lines, walls) or knot-like structures were observed in liquid crystals due to the presence of chiral dopants or special boundary conditions that do not allow for continuous strain in the entire arrangement[39,40]. Differently indexed topological defects are also known for graphene[41]. For the 2D self-assembly here, it is the combination of molecular shape and chirality that impose constraint at the supramolecular level. That is, the handedness and the azimuthal orientation of the t[4]HB molecules cause inevitably formation of topological defects.

Two-dimensional molecular self-assembly, as a model system for tiling, faces the challenge of poorly defined boundaries between molecules. As the overall lateral density approaches that of dense hexagonal packing, we consider nodes—whether voided or overcrowded—as integral to the tiling solution. Many approaches can be envisioned for translating this molecular assembly into rigid tiles. Although the basic tiles are only triangles, we present two possible solutions that account for chirality, symmetry at the nodes, and the triangle boundaries, as shown in Supplementary Fig. 11. A more extensive solution to the topological jigsaw tiling game based on one of such tiling sets is provided in Supplementary Fig. 12.

The Wang aperiodic tiling game employs sets of congruent square tiles, each possessing four numbered (or colored) edges[4]. Wang tiles were introduced as a tool for studying decidability in mathematical logic but are also important in theoretical computer science. The question of whether tiling exclusively by translation with matching edge numbers or colors—commonly referred to as the domino problem—possesses a periodic solution, has been proven to be undecidable[42]. However, recent research has demonstrated that a set of 11 Wang tiles is adequate for aperiodically tiling the plane[43]. In Penrose tiling, a set of shapes—typically two different types of tiles—are arranged according to specific rules, leading to a structure that covers a plane without periodic repetition but still exhibits long-range order. Several thin film systems have been reported to exhibit non-repetitive patterns, such as quasicrystallinity[44–48]. When applied at the molecular level, the challenge lies in guiding molecules to form such intricate patterns naturally through self-assembly. Recent molecular random

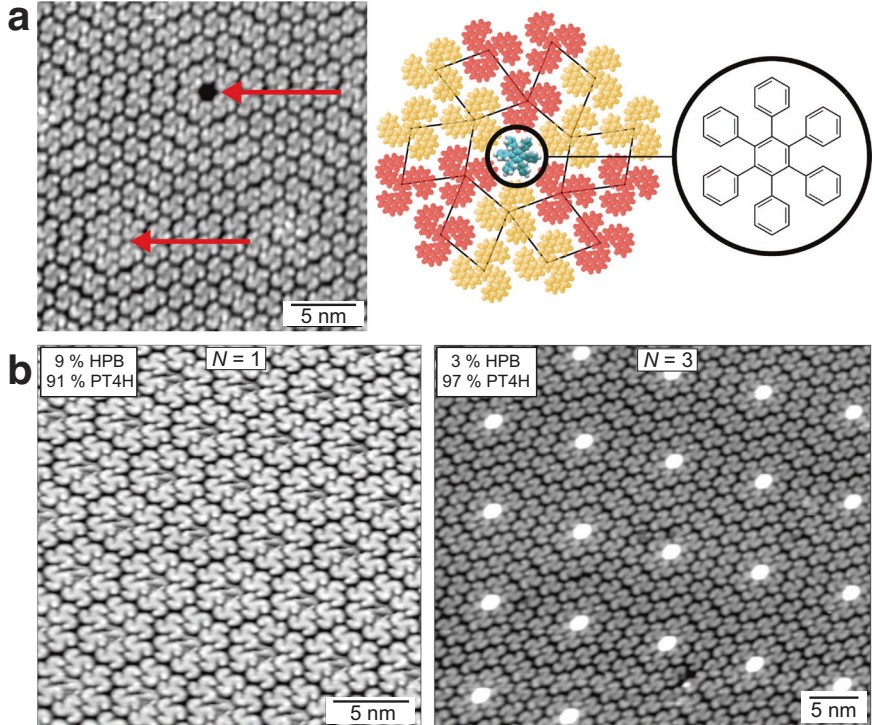

**Fig. 4 | Inhibiting entropy-driven self-assembly. a** Voids of zero nodes can be filled with impurities (red arrows). The molecular model shows a HPB molecule occupying a zero node. **b** STM images of t[4]HB monolayers modified with different amounts of HPB. A regular lattice of HPB, decorated with t[4]HB is established. The STM parameters were chosen such that the t[4]HB molecules are imaged best.

self-assemblies were based either on porous structures with directed polar intermolecular interactions[31–35,49,50] or had different species involved[51].

The here reported aperiodic tiling evolves from a single molecule that, due to its chirality, transmits into aperiodic tiling of triangular arranged ensembles. At the very least, our STM observations lend support to this conjecture, as only patterns lacking translational invariance have been observed. One of the fundamental rules of Wang tiling is the prohibition of tile reflection, which essentially transmits into the single handedness requirement of a solution to the monotile problem[7]. The original hat-shaped monotile ('einstein')[6], however, required that some of the tiles be reflected. This restriction persists here, as only a minority of opposite-handed molecules are present in all observed structures. While periodic solutions are conceivable (Supplementary Fig. 11), it remains unclear whether these consistencies enforce aperiodicity in the t[4]HB self-assembly on Ag(111). A mathematical proof demonstrating that subsets of the triangular tiles presented here inherently possess strictly aperiodic solutions is beyond the scope of this report.

In conclusion, the presented study demonstrates that aperiodic patterns may evolve via molecular 2D self-assembly on a nearly perfect surface. The crucial factors in this process are mobility, chirality, molecular flexibility and an aggregation scheme driven by the goal of maximizing entropy under topological constraint. The chirality-controlled interdigitation of the helical arms at the triangle boundaries prevents perfect size-matching of the edges, resulting in an offset determined by the molecule's handedness. This offset disrupts simple close packing, requiring the use of topological strategies to achieve dense packing. These strategies include three distinct topological defects and the ability to adjust the triangle size by one unit vector. Despite the defects, dense packing is still achieved, identical to the dense hexagonal packing within a triangle. As different solutions with identical packing densities are possible, entropy becomes an additional driving force. Enhancing entropy is achieved by adjusting the

minority content within a solid solution, facilitated by molecular adaptability. Starting from a chiral molecule, finally, the reported aperiodic tiling is formed. These results, along with detailed analysis, offer a set of design principles for future molecular monotiles, enabling dense aperiodic tiling through molecular self-assembly.

## Methods

### STM experiments and sample preparation
The experiments have been carried out in an ultrahigh vacuum (UHV) apparatus (base pressure $<5 \times 10^{-10}$ mbar) equipped with a variable-temperature scanning tunneling microscope (Aarhus SPM 150 Specs). Calibration of scan axes was deducted from atomic resolution STM of Cu(100). Noise reduction of images was achieved by moderate filtering (polynomial background subtraction). Drift correction, where applied, was achieved by shearing images until threefold symmetry of the t[4]HB self-assembled triangles was obtained.

The Ag(111) crystal surface was prepared by cycles of $Ar^+$ ion bombardment and subsequent annealing to 800 K. t[4]HB (see Materials below) was evaporated at 600 K as racemate from an effusion cell onto the Ag(111) surface held at room temperature. If not stated otherwise, STM images were acquired in constant current mode after cooling the sample to 120 K with liquid nitrogen.

### Computational modeling
Simulations were performed with AMBER force field molecular mechanics of the HyperChem 8.0 program. The surface/adsorbate systems were simulated by a four-layer Ag(111) slab with periodic boundary conditions. The Ag-atoms of the surface were kept fixed, but no constraint was applied to the molecules.

### Statistical analysis of enantiomeric excess
The enantiomeric excess of hexagonal phases fluctuates little in STM images of several hundred molecules. Therefore, the system is treated as a statistical ensemble. The standard error of the mean ($\hat{\sigma}$) from the

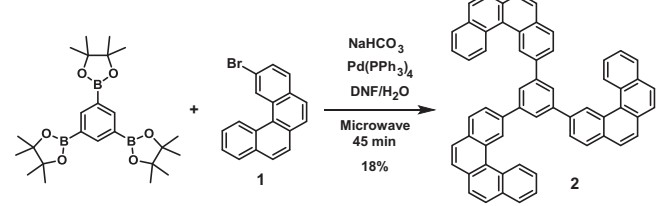

**Fig. 5 | Synthetic approach to t[4]HB.** t[4]HB (**2**) was synthesized from 1,3,5-tris(4,4,5,5-tetramethyl-1,3,2-dioxaborolan-2-yl)benzene and 2-bromobenzo[c]phenanthrene (**1**).

binomial distribution represents the errors of the observation mean. $\hat{\sigma}$ scales inversely with the sample size:

$$\hat{\sigma} = \sqrt{P_{minority} \times P_{majority} / n} \qquad (1)$$

and provides the standard deviation of the observational mean with respect to the population mean.

## Materials and instruments

All reagents and chemicals from commercial sources were used without further purification. 1,3,5-tris(4,4,5,5-tetramethyl-1,3,2-dioxaborolan-2-yl)benzene (TCI, 98%), 2-(bromomethyl)naphthalene (FLUOROCHEM, 98%), triphenylphosphine (ALFA AESAR, 99%), 4-bromobenzaldehyde (Sigma-Aldrich, 99%), tetrakis(triphenylphosphine)palladium(0) (Sigma-Aldrich, 99%). Solvents were dried and purified using standard techniques. Column chromatography was performed with analytical-grade solvents using Aldrich silica gel (technical grade, pore size 60 Å, 230–400 mesh particle size). Flexible plates ALUGRAM® Xtra SIL G UV254 from MACHEREY-NAGEL were used for TLC. Compounds were detected by UV irradiation (Bioblock Scientific) or staining with iodine, unless otherwise stated.

NMR spectra were recorded with a Bruker AVANCE III 300 ($^1$H, 300 MHz and $^{13}$C, 76 MHz) and Bruker AVANCE DRX 500 ($^1$H, 500 MHz and $^{13}$C, 125 MHz). Chemical shifts are given in ppm relative to tetramethylsilane (TMS) and coupling constants $J$ in Hz. Residual non-deuterated solvent was used as an internal standard.

Matrix-assisted laser desorption/ionization was performed on MALDI-TOF MS BIFLEX III Bruker Daltonics spectrometer using dithranol, DCTB, or α-terthiophene as matrix.

## Synthetic procedures

2-bromobenzo[c]phenanthrene (**1**):

Compound **1** has been synthesized in three steps from 2-(bromomethyl)naphthalene and 4-bromobezaldehyde according to the published method with 67% overall yield. The phosphonium bromide salt of 2-methylnaphthalene was first generated by reacting 2-(bromomethyl)naphthalene and triphenylphosphine, and then it was engaged in a classical Wittig coupling reaction with 4-bromobenzaldehyde. The 2-(4-bromostyryl)naphthalene intermediate thus obtained has been photocyclized in toluene in the presence of iodine and propylene oxide[25].

$^1$H NMR (300 MHz, chloroform-$d$) $\delta$ 9.29 (s, 1H), 9.05 (d, $J$ = 8.5 Hz, 1H), 8.03 (dd, $J$ = 6.4, 1.4 Hz, 1H), 7.92 (t, 2H), 7.86 (d, $J$ = 6.7 Hz, 3H), 7.78–7.62 (m, 3H).

The spectral data for this compound match those reported in ref. 52.

1,3,5-tris(benzo[c]phenanthren-2-yl)benzene (tris-[4]helicene)(**2**, t[4]HB):

In a 50 mL flask were dissolved 1,3,5-tris(4,4,5,5-tetramethyl-1,3,2-dioxaborolan-2-yl)benzene (50 mg, 0.11 mmol, 1 eq.), **1** (101.1 mg, 0.33 mmol, 3 eq.), NaHCO$_3$ (82.9 mg, 0.99 mmol, 9 eq.) and Pd(PPh$_3$)$_4$ (19 mg, 16.45 μmol, 0.15 eq.) in a 1/1 mixture of DMF/H$_2$O (5 mL).

The mixture was irradiated under micro-wave for 45 min at RT (Fig. 5). After completion, the organic layer was extracted with DCM, washed with H$_2$O, brine, dried over MgSO$_4$ and concentrated under vacuum. The crude was purified over preparative HPLC to afford 15 mg (18% yield) of t[4]HB as a white solid. Suitable crystals for X-ray analysis were obtained by slow evaporation in benzene.

$^1$H NMR (300 MHz, chloroform-$d$) $\delta$ 9.59 (s, 3H), 9.26 (d, $J$ = 8.3 Hz, 3H), 8.24 (s, 3H), 8.19 (d, $J$ = 8.3 Hz, 3H), 8.10 (s, 2H), 8.08–7.98 (m, 5H), 7.95 (d, $J$ = 8.8 Hz, 5H), 7.91–7.84 (m, 6H), 7.58–7.44 (m, 6H) (see Supplementary Fig. 13).

$^{13}$C NMR (125 MHz, chloroform-$d$) $\delta$ 143.34, 138.86, 133.68, 133.00, 131.50, 130.84, 130.43, 129.38, 128.70, 127.87, 127.84, 127.71, 127.36, 127.25, 126.98, 126.79, 126.62, 126.22, 126.12, 125.65 (see Supplementary Fig. 14).

MALDI-TOF = $m/z$ 756.2800; calculated = $m/z$ 756.2817.

## Crystallographic details

A single crystal of t[4]HB was mounted on a glass fiber loop using a viscous hydrocarbon oil to coat the crystal and then transferred directly to cold nitrogen stream for data collection. Data collection was performed at 150 K on an Agilent Supernova with Cu$K_\alpha$ ($\lambda$ = 1.54184 Å). The structure was solved by direct methods and refined on F$^2$ by full matrix least-squares techniques with SHELX programs (SHELXS-2013 and SHELXL-2016-2018)[53,54], using the WinGX graphical user interface[55].

**Single crystal X-ray structure of t[4]HB.** The t[4]HB crystallizes in the monoclinic system, centrosymmetric space group $P2_1$/c, with two independent helicene molecules in the unit cell and one and half independent crystallization benzene molecules (Supplementary Fig. 15). The conformation of the three [4]helicene units is $M,M,P$ for both independent molecules (Supplementary Figs. 15 and 16), the opposite configuration $P,P,M$ being generated through the inversion center. Intermolecular interactions of π–π stacking, suggested by Ph⋯Ph distances as short as 3.67 Å, sustain the supramolecular architecture in the solid state.

## Data availability

The data that support the findings of this study are available from Zenodo[56] and from the corresponding authors upon request. Crystallographic data for nine structures have been deposited with the Cambridge Crystallographic Data Centre (CCDC), deposition number CCDC 2092008 for **2**.

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

## Acknowledgements

Support by the Swiss National Science Foundation (Grants 212167, 182082, K.-H.E.; and grants 202775, 221265 C.W.) and the Grantová Agentura České Republiky - GAČR (24-11064S, K.-H.E.) is gratefully acknowledged. Financial support in France has been provided by the

CNRS, the University of Angers and the Région Pays de la Loire through the RFI LUMOMAT (grant to K.M.). K.-H.E. thanks Craig S. Kaplan for fruitful discussions.

## Author contributions

K.-H.E. and N.A. designed and supervised the project. K.M. and N.A. performed the chemical synthesis. J.V. and C.W. performed the experiments. J.V., M.B., C.W., K.M., N.A., and K.-H.E. analyzed the data. J.V., K.-H.E., and N.A. wrote the first draft. K.-H.E., M.B., C.W., and N.A. reviewed and edited the final draft.

## Competing interests

The authors declare no competing interests.
