## [Transparent Peer Review file · Nature Communications]

An aperiodic chiral tiling by topological molecular self-assembly

Corresponding Author: Professor Karl-Heinz Ernst

Version 0:

Reviewer comments:

Reviewer #1

(Remarks to the Author)

The manuscript has been improved.

The authors are presenting great STM images of molecular tilings.

They especially assess the origin of defects in the molecular ordering.

To my opinion the scheme in Supplementary Fig.3top deserves to be presented in the main manuscript,

it is essential to understand the molecular packing at the apex of the triangles to understand the different alignment of domain boundaries observed in the STM images.

I would suggest to remove "As different equal-density arrangements are possible, entropy maximization plays a crucial role" because this still remains speculative or the authors should soften the sentence with "The observation of different equal-density arrangements suggests that the entropy maximization may play a crucial role". The same should be done in the conclusion.

I nevertheless now support this paper for publication.

Reviewer #2

(Remarks to the Author)

Most of my concerns, as well as those raised by other reviewers, have been adequately addressed, leading to improved clarity in the revised manuscript. I recommend its acceptance by Nature Communications.

Reviewer #3

(Remarks to the Author)

I have considered carefully the combined comments of the four reviewers and the authors' responses to the reviewer comments.

Reviewer #1 commented that they are "not convinced that the reported tessellation is of high importance, nor is the main conclusion supported by the experimental data" and supported this comment with examples from the literature.

The authors provide a robust response to this criticism and I agree with the case that they make that there is a clear distinction between the close-packed adsorbate structure in the study under review and the rhombus tiling reported in reference 30 which is actually a porous self-assembled layer.

I agree with the assertion of the authors that the self-assembly behavior reported is distinct from previously reported studies. The actions taken by the authors to address the points made by Reviewer #1 are entirely appropriate and further improve the clarity of the paper.

Similarly the authors have made appropriate changes in light of the comments of Reviewer #2 and #3. With respect to some points, the authors have made no changes, but have provided a clear reasoning for rebuttal of the reviewers' comments.

My opinion is reinforced that this is an article of exceptionally high quality and I strongly recommend that it is accepted for

publication in Nature Communications.

Reviewer #4

(Remarks to the Author)

The ms was revised in a convincing way and this will be a really nice publication.

Small issues to be considered:

p.2: "aperiodic tiling driven solely by intermolecular forces" could be changed to "aperiodic tiling induced by intermolecular forces" as it can't be excluded that substrate registry and chemistry effects interfere in some way.

Response to reviewers' comments

REVIEWERS' COMMENTS

Reviewer #1 (Remarks to the Author):

The manuscript has been improved.

The authors are presenting great STM images of molecular tillings.

They especially assess the origin of defects in the molecular ordering.

To my opinion the scheme in Supplementary Fig.3^{top} deserves to be presented in the main manuscript, it is essential to understand the molecular packing at the apex of the triangles to understand the different alignment of domain boundaries observed in the STM images.

Action taken: Supplementary Fig. 3 has now become included as part of Figure 1. Accordingly, changes such as renumbering Supplementary Figs., text of figure caption and remarks in the text have been made.

I would suggest to remove "As different equal-density arrangements are possible, entropy maximization plays a crucial role" because this still remains speculative or the authors should soften the sentence with "The observation of different equal-density arrangements suggests that the entropy maximization may play a crucial role". The same should be done in the conclusion.

Reply: We disagree that entropy maximization is speculative as we have no other explanation.

Action taken: Nevertheless, we followed the suggestion of the reviewer and state now in the abstract: "The observation of different equal-density arrangements suggests that entropy maximization must play a crucial role." In the conclusion it's now: "Enhancing entropy is achieved by adjusting the minority content within a solid solution, facilitated by molecular adaptability instead of "Maximizing entropy is achieved by adjusting the minority content within a solid solution and is enabled by molecular flexibility."

I nevertheless now support this paper for publication.

Reviewer #2 (Remarks to the Author):

Most of my concerns, as well as those raised by other reviewers, have been adequately addressed, leading to improved clarity in the revised manuscript. I recommend its acceptance by Nature Communications.

Reviewer #3 (Remarks to the Author):

I have considered carefully the combined comments of the four reviewers and the

authors' responses to the reviewer comments.

Reviewer #1 commented that they are "not convinced that the reported tessellation is of high importance, nor is the main conclusion supported by the experimental data" and supported this comment with examples from the literature.

The authors provide a robust response to this criticism and I agree with the case that they make that there is a clear distinction between the close-packed adsorbate structure in the study under review and the rhombus tiling reported in reference 30 which is actually a porous self-assembled layer.

I agree with the assertion of the authors that the self-assembly behavior reported is distinct from previously reported studies. The actions taken by the authors to address the points made by Reviewer #1 are entirely appropriate and further improve the clarity of the paper.

Similarly the authors have made appropriate changes in light of the comments of Reviewer #2 and #3. With respect to some points, the authors have made no changes, but have provided a clear reasoning for rebuttal of the reviewers' comments.

My opinion is reinforced that this is an article of exceptionally high quality and I strongly recommend that it is accepted for publication in Nature Communications.

Reviewer #4 (Remarks to the Author):

The ms was revised in a convincing way and this will be a really nice publication.

Small issues to be considered:

p.2: "aperiodic tiling driven solely by intermolecular forces" could be changed to "aperiodic tiling induced by intermolecular forces" as it can't be excluded that substrate registry and chemistry effects interfere in some way.

Reply: We agree that surface registry might be involved.

Action taken: 'driven solely?' has been replaced by 'induced'

We thank all reviewers for their input and positive comments.